# Parenting Children with Autism Spectrum Disorder during Crises: Differential Responses between the Financial and the COVID-19 Pandemic Crisis

**DOI:** 10.3390/jcm11051264

**Published:** 2022-02-25

**Authors:** Katerina Papanikolaou, Vassiliki Ntre, Ioanna-Maria Gertsou, Evdokia Tagkouli, Chara Tzavara, Artemios Pehlivanidis, Gerasimos Kolaitis

**Affiliations:** 1Department of Child Psychiatry, Medical School, National and Kapodistrian University of Athens, Aghia Sophia Children’s Hospital, 11527 Athens, Greece; vassilikintre@yahoo.gr (V.N.); igertsou@gmail.com (I.-M.G.); etagkouli@gmail.com (E.T.); htzavara@med.uoa.gr (C.T.); gkolaitis@med.uoa.gr (G.K.); 2First Department of Psychiatry, Medical School, National and Kapodistrian University of Athens, Eginition Hospital, 11528 Athens, Greece; apechlib@med.uoa.gr

**Keywords:** Autism Spectrum Disorder, parenting stress, quality of life, coping strategies, depressive symptoms, COVID-19, economic crisis

## Abstract

Parenting a child with autism can be particularly stressful and challenging, especially during periods of crises. This study focuses on parenting children with autism during the COVID-19 pandemic in comparison to the economic crisis, six years ago. We administered the same set of questionnaires (CES-D, F-COPES, PSI-SF, and WHOQoL-BREF), along with a demographic characteristic and a COVID-19-related questionnaire to the same group of parents of children with autism as we did six years ago. Results indicated that during the COVID-19 crisis, the level of parenting stress and the distress due to personal factors related to demands of parenting, were significantly lower compared to the economic crisis, while the environment facet of quality of life was significantly higher. The depressive symptomatology was elevated during both periods. Finally, when addressing the pandemic crisis, parents were more likely to passively accept problematic issues, less able to obtain social support, and less able to acquire and accept help from others. In our sample, findings differentiate the genre of crises through the parents’ responses regarding the sense of competence in their parental role, their quality of life, and the coping strategies they implemented. The study implies that each crisis must be confronted with particular responses to particular needs.

## 1. Introduction

Autism Spectrum Disorder (ASD) is a neurodevelopmental disorder characterized by longstanding deficits in social communication along with stereotyped, restricted, and repetitive patterns of behaviors and interests. Parenting a child with ASD can be particularly stressful and challenging due to the particular characteristics of the disorder, the difficulty obtaining services, and the burden on the family’s finances [1,2,3,4]. Parents of children with ASD report low quality of life, increased parental stress, and more mental health issues compared to parents of children with typical development or other developmental disabilities [5,6,7,8,9,10,11,12,13,14]. In particular mothers, who are often reported to be the primary caregivers of children with ASD, are more vulnerable to psychological stress than fathers [2,15,16,17]. Social support, coping style, locus of control, self-efficacy, and positive family functions are among the factors that have been previously found to be associated with resilience in families of individuals with developmental disabilities and autism [18,19].

A crisis is a transitional period in which a person’s biopsychosocial integrity is temporarily disrupted or threatened [20]. The Family Adjustment and Adaptation Response (FAAR) models [21] describe crisis in a family as an imbalance arising from the number of demands exceeding the capacities of the family. More specifically, the crisis in mothers of people with ASD is reported to be composed of the combination of demands, internal capabilities, external resources, and subjective appraisal [22]. 

The COVID-19 pandemic is the most recent cause of a worldwide crisis. It resulted in unprecedented changes in daily lives that increased demands and has been a source of stress for millions of individuals and families across the world. In Greece during this period, all schools including special education schools were closed, and students were receiving tele-education. Even special treatment centers shut down, and children with ASD or other disabilities had to spend the whole day at home. Moreover, many companies had to suspend their operation resulting in a loss of income in many families. Changes in daily structure, attempts to mitigate lost opportunities for children’s learning and social interactions, increased screen time, and reduced physical activities are factors that have been discussed as important stressors for parents [23,24]. The population of children with disabilities, in particular autism, gained special attention; both opinion [25,26,27] and research papers discussed the possible impact of the COVID-19 pandemic on individuals with ASD [28,29,30,31,32,33] and their families [28,31,32,34,35]. They suggested strategies for them to better accommodate to the situation. Parents of individuals with ASD were more likely to have mental health problems compared to parents whose children had an intellectual disability, a visual or a hearing impairment [36], or parents of typically developing youths [37]. They reported high levels of psychological distress and disruption in their lives during the pandemic [31,34,37,38,39,40,41,42]. Some possible positive aspects of experiences during the pandemic have been also reported by parents of typically developing children [43,44] and caregivers of children with ASD [28,32,35,44,45,46]. These include the elimination of many daily pressures [32,45]; creating a more relaxing environment [28]; having more time to teach new skills, such as autonomy or house care-related skills [46]; having opportunities for increased family interaction and strengthening relationships [35]; developing positive characteristics such as appreciation, gratitude, and tolerance; and finding new hobbies [44]. 

When the COVID-19 pandemic emerged, Greek parents were in the unpleasant position of already having experienced the devastating impact of a recent financial crisis. Even before the emergence of the pandemic, the economic crisis had depleted Greek people’s psychological and financial resources while public health services deteriorated [47]. Several reports showed that depression and suicide increased in adults during the period of the financial crisis [48,49,50,51]. Child and adolescent mental health services underwent budget cuts leading to some services not being fully operational. Many non-profit child and adolescent mental health centers and psychosocial rehabilitation units, including establishments specialized for autism, also closed during the economic crisis. In addition, parents had to discontinue their children’s treatments because of their inability to cover by their own means the expenses the treatments entail [52]. 

The aim of the present study was to assess the parental stress, quality of life, coping strategies, and presence of depressive symptoms in a cohort of parents, mainly mothers, of children with ASD during the first phase of the COVID-19 crisis. It also aimed to compare our findings to the responses that the same cohort of parents gave to the same set of questionnaires during the financial crisis, approximately six years ago. 

## 2. Method

### Participants and Procedure

During the years 2014–2015, in the heart of the economic crisis, 62 parents of children with ASD completed a set of questionnaires on parenting stress, depressive symptomatology, quality of life, and coping strategies, participating in a research project on the needs, burdens, quality of life, and coping strategies of families with children with ASD [2,3]. The primary caregiver was asked to complete the questionnaires. All parents were attending the ASD Outpatient Clinic of a University Department and constituted consecutive cases coming for follow-up assessments. Their children or adolescents had received an ASD diagnosis according to DSM-5 criteria [53] in the past and had undergone a standard psychometric evaluation. Exclusion criterion was the inability of parents to read or write adequately in Greek.

During the first phase of the COVID-19 pandemic, two months after the implementation of restrictive measures by the Greek Government, the same parents were asked to respond to a web-based survey, including a questionnaire on demographics and other characteristics of the sample (Table 1) and the same battery of instruments they had completed during the economic crisis period. 

Out of 62 parents, 50 mothers and six fathers agreed to participate in the survey. The DSM-5 level of functioning of the youngsters during the two periods remained the same except for the case of one subject, whose level of functioning deteriorated from level 2 to level 3.

## 3. Instruments

***Demographic Characteristics and COVID-19 related Questionnaire.*** This questionnaire was developed for the needs of the current study and covers the caregivers’ and children’s demographic characteristics and information on the impact of the COVID-19 pandemic on the family (Table 1).

***Center for Epidemiologic Studies Depression Scale (CES-D)*** [54]. This is a short self-reporting scale designed to measure depressive symptoms in the general population. It consists of 20 items, measuring the presence of depressive symptoms in the past week, on a four-point scale ranging from 0 (rarely) to 3 (most of the time). The possible range of scores is zero to 60. A score of 16 or higher indicates risk of clinical depression. It has been validated in the Greek population, showing a very good internal consistency (Cronbach’s alpha: 0.95) [55,56].

***Family Crisis Oriented Personal Scales (F-COPES)*** [57]. It is a 30-item, self-reporting instrument measuring a family’s coping style. It includes two levels of interaction: (a) the ways a family internally handles difficult situations and problems and (b) the ways the family externally handles problems that emerge outside its boundaries. The responses range from 1 (strongly agree) to 5 (strongly disagree) and produce a total score and five subscales scores referring to: (a) acquiring social support, (b) reframing, (c) seeking spiritual support, (d) mobilizing family to acquire and accept help, and (e) passive appraisal. It is a reliable and valid tool that measures coping strategies and level of adaptation. The Greek version 0f F-COPES has shown a Cronbach’s alpha of 0.86 and a test–retest reliability of 0.81 [58]. 

***Parenting Stress Index Short-Form (PSI-SF)*** [59]. It consists of 36 items deriving from the Parenting Stress Index [60] and is a measure of the stresses that a parent is experiencing in his/her role. It comprises three subscales labeled Parental Distress (PD), Parent-Child Dysfunctional Interaction (PCDI), and Difficult Child (DC). The PD subscale measures the level of distress due to personal factors related to demands of parenting; the PCDI subscale assesses parents’ dissatisfaction with their interactions with their children, and the DC subscale measures parents’ perceptions of the characteristics of their child’s behavior and how difficult it can be to manage. It also includes a Defensive Responding scale indicating parents’ denial or minimization of problems. Participants use a 5-point Likert scale indicating the degree to which they agree with each statement. The Total Stress score is a composite score of the subscale scores. In our study, PSI-SF showed a very good internal consistency both for the total scale (Cronbach’s α = 0.91) and all subscales (0.91: Parenting Distress; 0.82: Parent–Child Dysfunctional Interaction; 0.73: Difficult Child; 0.85: Defensive Responding).

***World Health Organization Quality of Life-BREF (WHOQoL-BREF)*** [61]. The WHOQOL-BREF is an abbreviated version of WHOQOL-100, developed by the World Health Organization [62]. WHOQOL-BREF consists of 24 items corresponding to 24 QOL (thematic) facets, and two items comprising an overall quality of life/general health facet. Items are organized into four domains: (1) physical health, (2) psychological health, (3) social relationships, and (4) environment. The WHOQOL-BREF Greek version has demonstrated satisfactory psychometric properties [63].

## 4. Statistical Analysis

Quantitative variables were expressed as mean (Standard Deviation) or as median (interquartile range). Qualitative variables were expressed as absolute and relative frequencies. Paired students’ *t*-tests were used for comparisons in all study scales between measurements occurring during the economic and pandemic crisis. The Benjamini-Hochberg procedure was applied to control the false discovery rate for independent tests. In order to examine the differences in all study scales between measurements after adjusting for parental educational level, income after the pandemic, DSM-5, and child’s age, repeated measures ANOVA were conducted. Pearson’s correlations coefficients were used to explore the association of all understudy scales during the pandemic crisis. Multiple linear regression was conducted, in a stepwise method (p for entry 0.05, p for removal 0.10) to find factors associated with the total PSI score. As independent variables, sample characteristics as well as F-COPES subscales were used. All reported *p* values are two-tailed. Statistical significance was set at *p* < 0.05 and analyses were conducted using SPSS statistical software (version 22.0).

## 5. Results

The sample consisted of 56 parents with mean age 49.6 years (SD = 7.8) (Table 1 presents sample characteristics). Two PSI subscales, namely Defensive Responding and Parental Distress scores, along with Total PSI score were significantly decreased from the financial to the COVID-19 crisis. Significant reduction in Mobilizing Family to Acquire and Accept Help and Acquiring Social Support subscales was recorded. Moreover, scores on Passive Appraisal had a significant increase. No changes in CES-D scale were recorded from the financial to the COVID-19 crisis. As far as WHO-QOL subscales were concerned, a significant increase in the mean score of the Environment subscale was found (Table 2). Using the Benjamini-Hochberg procedure, all aforementioned significant comparisons remain statistically significant.

After having adjusted for parental educational level, income after the pandemic, DSM-5, and the child’s age, it was found that Defensive Responding score remained significantly lower during the COVID-19 crisis (*p* < 0.001), as did Parental Distress score (*p* = 0.007) and total PSI score (*p* = 0.005). Moreover, Mobilizing Family to Acquire and Accept Help and Acquiring Social Support subscales were significantly lower during the COVID-19 crisis (*p* = 0.002 for both subscale), after adjusting for parental educational level, income after the pandemic, DSM-5, and the child’s age. On the contrary, Passive Appraisal and Environment subscales were significantly greater during the COVID-19 crisis after adjusting for parental educational level, income after the pandemic, DSM-5, and the child’s age (*p* < 0.001 and *p* = 0.001, respectively).

Pearson’s correlation coefficients among all scales are presented in Table 3. In general, greater stress was significantly associated with greater depression, worse coping, and worse quality of life. Moreover, greater depression was significantly associated with worse coping and worse quality of life, while better coping was associated significantly with better quality of life.

When multiple linear regression was conducted it was found that Passive Appraisal (β = −3.17; SE = 1.25; *p* = 0.015) and Acquiring Social Support β = −1.16; SE = 0.52; and *p* = 0.032) were significantly associated with total PSI score.

## 6. Discussion

In the present study, we compared the impact of the two different social crises, the financial and the COVID-19 crises, on the parenting stress, depressive symptomatology, quality of life, and coping strategies of mothers who have children and adolescents with ASD.

### 6.1. Parenting Stress

Parental Stress as measured in PSI-SF refers to the stresses that a parent is experiencing in his/her role; it does not include all life stresses, just the stress deriving from his/her parenting. In particular, Parental Distress measures the sense of parenting competence, the stress associated with restrictions in life, conflict with the other parent, social support, and depression. As expected from previous findings in a Greek sample [2], parental stress was significantly associated with greater depression, worse coping, and worse quality of life. Moreover, two dysfunctional coping mechanisms, namely increased passive appraisal and reduced social support, predicted elevated parental stress during the pandemic.

Interestingly the parenting stress during the COVID-19 crisis as reflected in the total PSI score and the Parental Distress Score was significantly lower compared to the parenting stress during the economic crisis. Moreover, during the pandemic, mothers made fewer attempts to deny or minimize problems when responding to the questionnaire.

Differences between the two types of crises regarding the impact on daily life and the strategies to accommodate to the change might explain differences in the parental distress measures.

The financial crisis in Greece has been a prolonged and sustained economic downturn. The Greek Government had to implement austerity measures that had various consequences on the daily lives of Greeks. The funding for mental health was reduced. Many parents lost their jobs or had their salaries dramatically reduced resulting in the discontinuation of their children’s treatments. Moreover, parents of both typically developing children and children with autism or other developmental disorders tend to spend a lot of money on their children’s after-school activities [2]. They had, therefore, to face a twofold difficult situation that could affect their sense of competence as parents: first, their children had to discontinue both their treatment and their extra activities because of their parents’ inability to cover expenses, and second, they had to be away from home to find extra financial resources to cover basic family needs.

During the COVID-19 crisis and the restrictions implemented by the Government to control the pandemic, the daily lives of Greeks changed as well, but in a different direction. Approximately 68% of children and adolescents in our study stopped receiving any kind of intervention. Contrary to the economic crisis, parents were spending most of their time at home because of tele-working, because they were entitled to claim special time-off from their work to stay with their children, or because they were temporarily out of work. They lost the support of grandparents whose role in the Greek family is central. However, they had the opportunity to be with their children and strengthen their relationships. Moreover, fathers who are often reported to be less involved in the care of their child with ASD [2] had the chance to spend more time with their child and support the mother. Unfortunately, in our study, 35.8% of the parents reported a decrease in the income of the family. Nevertheless, there was a halting of the continuous race of getting involved in numerous and all sorts of costly activities, which resulted in a decrease of expenses. Moreover, the crisis was a worldwide and pan-European phenomenon, and Greece received financial rescue funds that supported many of these families. This difficult situation was, therefore, probably conceived as a temporary and universal phenomenon imposed by governmental restrictions, not related to parents’ competence to fulfill their role.

During our clinical practice and in the literature, there have been many reports about difficult behaviors of children with ASD during the COVID-19 crisis [29,41]. In our study, 10.7% of the families had to increase the dose of a previously prescribed sedative for their child. Overall, though, it seems that during the first phase of the pandemic crisis the primary caregivers of people with ASD in Greece felt more competent, less restricted, and more supported in their role as a parent compared to the economic crisis period.

### 6.2. Depressive Symptomatology

Parents of disabled children, and especially parents of children with autism, experience more challenges and are likely to report mental health problems compared to parents of typically developing children [2,64,65]. In particular mothers, as the primary caregivers of children with autism [2,16], report higher levels of psychological stress [15,17]. In accordance with previous findings [2], increased symptoms of depression were significantly associated with more dysfunctional coping strategies and worse quality of life in our study.

Mothers’ depressive symptomatology as reflected in CES-D scoring, showed no difference between the first phase of the COVID-19 and the financial crisis period. In both cases, the CES-D score was elevated. The negative impact of the economic crisis on the depressive symptomatology and wellbeing of the general population has been previously studied [48,50,66] although not in particular in the population of parents with individuals with ASD. Financial difficulties and burdens have been repeatedly reported, though, as important stressors for those parents [3,64]. High levels of depression in parents of disabled children have been described during the COVID-19 pandemic crisis [34,36,67]. It seems that no matter the nature of the crisis, the threatening of an individual’s functioning, values, and goals during this period and the imbalance between demands, family capabilities, external resources, and subjective appraisal that has been described to characterize crises in families of individuals with ASD [22], appears to have an equally strong impact on parents’ levels of depression.

### 6.3. Quality of Life

Quality of life is a multidimensional concept that incorporates physical health, psychological state, social relationships, and relationships with salient features of the environment [68]. In line with a previous study in a Greek sample of mothers of children with ASD [2], a significant association was found between quality of life and coping mechanisms.

If we compare the two crises periods, no significant change was found in the first three domains, while a significant increase in the Environment subscale score was noted during the COVID-19 pandemic. The environment domain incorporates facets such as financial resources, opportunities for acquiring information, physical safety and security, accessibility to health services, transport, participation in leisure activities, and home environment. Although at first glance one would expect all these areas to be seriously affected by the pandemic crisis, it seems that during the economic crisis, the impact was even higher. There might be many explanations for this finding: as mentioned before, during the pandemic crisis the financial rescue funds alleviated many families from the financial strain. Moreover, information on the new coronavirus, which was the main health issue at the time, was available everywhere. Despite the fear of getting a COVID-19 infection, at that point the situation in Greece was quite satisfactory and staying at home gave a sense of safety. Access to health services was eliminated to emergencies but services did not shut down as they did during the financial crisis. Tele-medicine was available in many instances, and there was no need for transportation. There were no opportunities for outdoor leisure activities, but parents had more time to pursue hobbies at home. Finally, the home environment with social distancing and all kinds of tele-activities can be an autism-friendly environment and less stressful for both people with ASD and some parents with autism broader phenotype features.

### 6.4. Coping Strategies

Coping consists of “cognitive and behavioral efforts to manage psychological stress” [69]. Both problem-focused (e.g., reappraisal, reframing) and emotion-focused (e.g., social support, spirituality, and respite) coping strategies are used by the parents of individuals with ASD [2,70]. Among the five coping strategies assessed by the F-COPES scales, three strategies showed a significant change during the pandemic crisis compared to the financial one. Lower scores in Mobilizing Family to Acquire and Accept Help and Acquiring Social Support probably reflect the social distancing and other restrictive measures implemented by the government to control the COVID-19 pandemic. Access to resources and social networks was reduced. Families lost contact with relatives, family friends, and most importantly with grandparents, which in Greece usually spend long hours with their grandchildren and support families. At the same time, scores in Passive Appraisal increased. Parents felt less confident in their ability to alter the outcome of a health crisis compared to a financial one. It is probably to be expected that the locus of control when fighting a pandemic is more external than when facing an economic crisis.

### 6.5. Limitations of the Study

One important limitation of the present study is that there has been a period of six years between the two crises. We cannot exclude the possibility that the differences between the results in the two time periods are related to reasons other than the economic or COVID-19 crisis. Although in our sample the DSM-5 level of functioning of the youngsters remained almost the same, there is evidence that as the individuals with ASD become older their behavioral problems sometimes worsen. Parents have to face stressors associated with transitions across key developmental periods [71,72]. On the contrary, parents obtain the skills to manage problematic situations more effectively. Findings on the impact of children’s age on the wellbeing of their parents have been inconsistent. Some studies revealed higher burden in parents of younger children [71,73,74,75], others suggest increasing stress with age [72,76], while others indicate no influence of age on parenting stress [11]. Although other factors might be important, we cannot ignore the fact that both crises had a perverse impact on the lives of Greek people and both periods were heavily characterized by this impact. Apart from being a limitation, the six years distance between the two crises can also be considered as an advantage since there was less of an overlap between the two periods. Another limitation of the study is the fact that we do not have data on the same cohort during a non-crisis period. Therefore, we can only extrapolate conclusions about differences in parenting during different types of crises and not about the parenting of these families under ordinary conditions. We must also consider the fact that our results apply mainly to mothers, who in the vast majority of cases were the main caregiver of their child. As such, we cannot conclude how different the impact is on fathers. In addition, our sample was not very large, nevertheless, it was a within-subjects comparison, which increases the rigorousness of our findings. Sampling biases should also be considered. The study’s results depended on participants attending a particular ASD outpatient clinic, limiting the wide-range effects of the pandemic on families of children with ASD throughout the country. Finally, our data refer to the first two months of the “stay at home” period of the COVID-19 crisis. It will be very interesting to study the possible additive effects of the prolonged crisis as well as its long-term influence on the wellbeing and coping strategies of parents of individuals with ASD.

## 7. Conclusions

The findings of the present study indicate that the financial and pandemic crises might elicit different responses from the primary caregivers of individuals with ASD. In our sample, the nature of crisis influenced the sense of competence in their role as parents, facets of their quality of life, and the coping strategies they implemented. In line with findings from parents of both typically developing children and children with ASD, it seems that the lockdown was not a uniformly negative experience for the parents of children with ASD.

The differential responses imply, though, that particular needs have to be addressed during different types of crises. In our sample parents’ sense of competence in their parental role, safety issues, accessibility to services, or participation in leisure were more influenced during an economic crisis. Maintaining access to social networks and adequate resources, and encouraging an internal locus of control, seemed to be more important during the physical and social distance period. Even during “ordinary” periods, factors such as social support, coping style, locus of control, self-efficacy, and positive family functions impact the resilience of families raising a child with developmental disabilities or autism. In times of crises, it seems even more important to turn our lenses to caregivers in addition to focusing on individuals with ASD and addressing their particular needs so as to enhance their resilience and ability to handle adversities.

## Figures and Tables

**Table 1 jcm-11-01264-t001:** Sample characteristics.

	N (%)
Gender	
Mother	50 (89.3)
Father	6 (10.7)
Age, mean (SD)	49.6 (7.8)
Educational status	
Middle/High school	16 (28.6)
University/Post-graduate studies	40 (71.4)
Married	43 (76.8)
Number of children, median (IQR)	2 (2.2)
Child’s age, mean (SD)	17.3 (3.3)
Child’s gender	
Boys	44 (78.6)
Girls	12 (21.4)
Income after pandemic	
Increased	1 (1.8)
Stable	35 (62.5)
Decreased	17 (30.4)
Nullified	3 (5.4)
Recent use of sedatives/psychotropic drugs by the mother	12 (21.4)
Need of child’s dosage increase during the pandemic	6 (10.7)
Ask for additional psychological support or help	
Not at all	46 (82.1)
A little	5 (8.9)
Moderately	3 (5.4)
Much	1 (1.8)
Very much	1 (1.8)
Child’s treatment continued	
No	36 (67.9)
Yes, with physical presence of the therapist	3 (5.4)
Yes, remotely	15 (26.8)
Wisc III score, mean (SD)	78 (25.6)
DSM-5—level of functioning	
1	15 (26.8)
2	20 (35.7)
3	21 (37.5)

**Table 2 jcm-11-01264-t002:** Changes in PSI, F-COPES, CES-D, and WhoQolBref subscales from the financial to the COVID-19 crisis.

	Financial Crisis		COVID-19 Crisis		Change		*p* Paired *t*-Test
	Mean	SD	Mean	SD	Mean	SD	
**PSI**							
Defensive Responding	18.53	4.36	16.02	5.26	−2.51	4.77	<0.001
Parental Distress	34.25	8.46	29.96	10.02	−4.29	9.60	0.003
Parent-Child Dysfunctional Interaction	29.73	8.20	27.91	7.83	−1.82	7.33	0.057
Difficult Child	34.73	9.84	34.11	7.69	−0.62	9.81	0.484
Total PSI score	119.10	25.87	108.00	26.22	−11.10	23.42	<0.001
**FCOPES**							
Reframing	30.27	4.13	30.09	5.36	−0.18	6.36	0.787
Mobilizing Family to Acquire and Accept Help	15.58	2.65	14.27	2.94	−1.31	3.35	0.008
Seeking Spiritual Support	10.71	3.71	11.39	4.31	0.69	4.07	0.291
Acquiring Social Support	26.08	7.08	22.84	6.77	−3.24	9.13	0.007
Passive Appraisal	10.22	3.13	14.18	2.95	3.96	5.01	<0.001
Overall FCOPES	93.35	13.15	95.75	13.16	2.40	18.38	0.523
**CES-D**	17.90	11.95	17.93	11.97	0.02	12.72	0.845
**WhoQolBref**							
Overall health	13.39	3.06	13.36	2.78	−0.03	3.40	0.802
Physical health	14.45	2.19	14.27	2.51	−0.18	2.36	0.513
Psychological health	13.29	2.52	13.39	1.83	0.10	2.51	0.910
Social relationships	12.60	2.85	13.19	2.70	0.58	2.98	0.271
Environment	11.56	2.11	12.81	1.90	1.25	2.21	<0.001

**Table 3 jcm-11-01264-t003:** Pearson’s correlation coefficients among all understudy scales during the COVID-19 crisis.

	1	2	3	4	5	6	7	8	9	10	11	12	13	14	15	16	17
1. Defensive Responding	1.00																
2. Parental Distress	0.92 ***	1.00															
3. Parent-Child Dysfunctional Interaction	0.68 ***	0.70 ***	1.00														
4. Difficult Child	0.41 **	0.40 ***	0.67	1.00													
5. Total PSI score	0.88 ***	0.89 ***	0.90 ***	0.73 ***	1.00												
6. Reframing	−0.21	−0.20	−0.29 *	−0.22	−0.27 *	1.00											
7. Mobilizing Family to Acquire and Accept Help	−0.23	−0.25	−0.04	−0.02	−0.16	0.26	1.00										
8. Seeking Spiritual Support	−0.03	0.00	0.03	0.10	0.04	0.11	0.22	1.00									
9. Acquiring Social Support	−0.21	−0.32 *	−0.24	−0.17	−0.29 *	0.28 *	0.36 **	0.01	1.00								
10. Passive Appraisal	−0.20	−0.19	−0.27 *	−0.29 *	−0.28 *	−0.10	0.09	−0.12	0.01	1.00							
11. Overall F Copes	−0.33 *	−0.36 **	−0.33 *	−0.23	−0.37 **	0.65 ***	0.61 ***	0.41 **	0.75 ***	0.18	1.00						
12. CES-D	0.67 ***	0.69 ***	0.38 **	0.28 *	0.60 ***	−0.30 *	−0.23	−0.07	−0.35 **	−0.10	−0.44 **	1.00					
13. Overall health	−0.51 ***	−0.49 ***	−0.38 **	−0.19	−0.46 ***	0.38 **	0.27 *	0.08	0.23	0.12	0.40 **	−0.43 **	1.00				
14. Physical health	−0.50 ***	−0.45 ***	−0.39 **	−0.21	−0.45 ***	0.56 ***	0.25	0.17	0.20	0.07	0.51 ***	−0.72 ***	0.54 ***	1.00			
15. Psychological health	−0.64 ***	−0.67 ***	−0.54 ***	−0.27 *	−0.62 ***	0.36 **	0.24	0.12	0.42 **	0.11	0.54 ***	−0.67 ***	0.43 **	0.76 ***	1.00		
16. Social relationships	−0.64 ***	−0.65 ***	−0.52 ***	−0.26 *	−0.61 ***	0.39 **	0.37 **	0.28 *	0.42 **	0.28 *	0.64 ***	−0.68 ***	0.63 ***	0.65 ***	0.61 ***	1.00	
17. Environment	−0.31 *	−0.37 **	−0.26	−0.11	−0.31 *	0.43 **	0.23	0.04	0.22	0.22	0.43 **	−0.53 ***	0.47 ***	0.62 ***	0.56 ***	0.49 ***	1.00

* *p* < 0.05; ** *p* < 0.01; *** *p* < 0.001.

## Data Availability

The data presented in this study are available on request from the corresponding author.

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
