# Peer review of "Parenting Children with Autism Spectrum Disorder during Crises: Differential Responses between the Financial and the COVID-19 Pandemic Crisis"

_jcm, 2022, doi:10.3390/jcm11051264_

Round 1
Reviewer 1 Report
It’s a hard work to do a six-year follow-up study. As I have mentioned that it is really interesting to explore the responses of parents of children with ASD in different crisis, and I think the topic of the study is very interesting. The data about the characteristics of the participants and their families at the two data collection time points was added in the revised manuscript. But in my opinion, the statistical analysis methods need to be improved, and regression analysis is necessary. The confounding factors are not well controlled, so the credibility of the study conclusion will be greatly affected.
The aim of this study is not very clear. There were four variables (CES-D, F-COPES, PSI-SF and WHOQoL-BREF) in the study, but the relationship among the variables is not clear, and there is no theory to explain their relationship._
The results are not well organized. It is suggested to combine Table 2 to Table 5, and I suggest to focus on one or two important dependent variables, and compare their changes at the two crisis time points. Controlling the potential confoundings are necessary.
Author Response
Thank you for your helpful comments. Below we are explaining the changes we made in our manuscript according to your suggestions.
“But in my opinion, the statistical analysis methods need to be improved, and regression analysis is necessary. The confounding factors are not well controlled, so the credibility of the study conclusion will be greatly affected.”
A regression analysis for total PSI score that had a significant change during follow up and also a new analysis controlling for confounding factors, were performed. The results were added in the correspondence section.
“There were four variables (CES-D, F-COPES, PSI-SF and WHOQoL-BREF) in the study, but the relationship among the variables is not clear, and there is no theory to explain their relationship”
An additional analysis with the correlation of the variables of interest was added in the results section.
“It is suggested to combine Table 2 to Table 5, and I suggest to focus on one or two important dependent variables, and compare their changes at the two crisis time points. Controlling the potential confoundings are necessary.”
Table 2 to Table 5 were merged. We have added the results of a repeated measurements analysis of variance controlling for child's age, income, DSM-5 –level of functioning and educational status; for variables that showed a significant change during follow up. A regression analysis for PSI was also performed.
Thank you for your helpful comments. Below we are explaining the changes we made in our manuscript according to your suggestions.
“But in my opinion, the statistical analysis methods need to be improved, and regression analysis is necessary. The confounding factors are not well controlled, so the credibility of the study conclusion will be greatly affected.”
A regression analysis for total PSI score that had a significant change during follow up and also a new analysis controlling for confounding factors, were performed. The results were added in the correspondence section.
“There were four variables (CES-D, F-COPES, PSI-SF and WHOQoL-BREF) in the study, but the relationship among the variables is not clear, and there is no theory to explain their relationship”
An additional analysis with the correlation of the variables of interest was added in the results section.
“It is suggested to combine Table 2 to Table 5, and I suggest to focus on one or two important dependent variables, and compare their changes at the two crisis time points. Controlling the potential confoundings are necessary.”
Table 2 to Table 5 were merged. We have added the results of a repeated measurements analysis of variance controlling for child's age, income, DSM-5 –level of functioning and educational status; for variables that showed a significant change during follow up. A regression analysis for PSI was also performed.
Thank you for your helpful comments. Below we are explaining the changes we made in our manuscript according to your suggestions.
“But in my opinion, the statistical analysis methods need to be improved, and regression analysis is necessary. The confounding factors are not well controlled, so the credibility of the study conclusion will be greatly affected.”
A regression analysis for total PSI score that had a significant change during follow up and also a new analysis controlling for confounding factors, were performed. The results were added in the correspondence section.
“There were four variables (CES-D, F-COPES, PSI-SF and WHOQoL-BREF) in the study, but the relationship among the variables is not clear, and there is no theory to explain their relationship”
An additional analysis with the correlation of the variables of interest was added in the results section.
“It is suggested to combine Table 2 to Table 5, and I suggest to focus on one or two important dependent variables, and compare their changes at the two crisis time points. Controlling the potential confoundings are necessary.”
Table 2 to Table 5 were merged. We have added the results of a repeated measurements analysis of variance controlling for child's age, income, DSM-5 –level of functioning and educational status; for variables that showed a significant change during follow up. A regression analysis for PSI was also performed.
Thank you for your helpful comments. Below we are explaining the changes we made in our manuscript according to your suggestions.
“But in my opinion, the statistical analysis methods need to be improved, and regression analysis is necessary. The confounding factors are not well controlled, so the credibility of the study conclusion will be greatly affected.”
A regression analysis for total PSI score that had a significant change during follow up and also a new analysis controlling for confounding factors, were performed. The results were added in the correspondence section.
“There were four variables (CES-D, F-COPES, PSI-SF and WHOQoL-BREF) in the study, but the relationship among the variables is not clear, and there is no theory to explain their relationship”
An additional analysis with the correlation of the variables of interest was added in the results section.
“It is suggested to combine Table 2 to Table 5, and I suggest to focus on one or two important dependent variables, and compare their changes at the two crisis time points. Controlling the potential confoundings are necessary.”
Table 2 to Table 5 were merged. We have added the results of a repeated measurements analysis of variance controlling for child's age, income, DSM-5 –level of functioning and educational status; for variables that showed a significant change during follow up. A regression analysis for PSI was also performed.
Thank you for your helpful comments. Below we are explaining the changes we made in our manuscript according to your suggestions.
“But in my opinion, the statistical analysis methods need to be improved, and regression analysis is necessary. The confounding factors are not well controlled, so the credibility of the study conclusion will be greatly affected.”
A regression analysis for total PSI score that had a significant change during follow up and also a new analysis controlling for confounding factors, were performed. The results were added in the correspondence section.
“There were four variables (CES-D, F-COPES, PSI-SF and WHOQoL-BREF) in the study, but the relationship among the variables is not clear, and there is no theory to explain their relationship”
An additional analysis with the correlation of the variables of interest was added in the results section.
“It is suggested to combine Table 2 to Table 5, and I suggest to focus on one or two important dependent variables, and compare their changes at the two crisis time points. Controlling the potential confoundings are necessary.”
Table 2 to Table 5 were merged. We have added the results of a repeated measurements analysis of variance controlling for child's age, income, DSM-5 –level of functioning and educational status; for variables that showed a significant change during follow up. A regression analysis for PSI was also performed.
Reviewer 2 Report
Interesting paper. Small sample size, but sufficient to understand the main idea. Unfortunately, both crisp had and will have impact on the most vulnerable and the main concern is that the study was conducted ti early regrading the pandemic situation...I suggest the authors to replicate the research two years later.
Author Response
Thank you for your comments. We have already collected data during the second year of the quarantine and we intend to collect more data next year.
Reviewer 3 Report
The study was conducted within the framework of quasi-experimental design “a series of time samples” (D.T. Campbell). However, the correctness of the design is violated by the lack of measurements in the absence of crises. Of course, this fact raises doubts about the validity of the results of the study. Nevertheless, in the section “Limitations of the study” the authors present acceptable arguments in defense of the plausibility of the results obtained.
Abstract.
A brief description of the results should be made not in terms of measurements (variables), but in terms of constructs. And it is inappropriate to give data about p-level in the abstract.
Introduction.
Often some taking points are supported by references to 7 to 10 sources, as if all these publications are about the same thing (lines 55 - 57, 59 - 63). It is necessary to explain in more details exactly what ideas related to this study are contained in these sources. For example, in line 57 there are links to 7 sources (Some possible positive aspects of experiences during the pandemic have also been reported by parents of typically developing children [43-44] and caregivers of children with ASD [26,43,31,40,41,45,46]). And then the detailing of these taking points follows, but without references to sources (lines 63-67). Apparently, the references listed above should be connected with specific details in lines 63-67.
Instruments.
There is no data on the Greek adaptation of Parenting Stress Index Short-Form (PSI-SF).
Results.
The tables show Statistical significance without taking into account Multiple comparisons problem. It would be necessary to introduce Multiple testing correction, for example, using the Holm or Benjamini-Hochberg. Even though it obviously won't affect the results significantly.
References
Violation of numbering in the list of references – double numbering.
Lines 315 - 319 - Table 3 is shown twice.
The language needs extensive proofreading.
Recommend for publication, taking into account the comments and the suggestions made.

Author Response
Thank you for your helpful comments.
Abstract.
“A brief description of the results should be made not in terms of measurements (variables), but in terms of constructs. And it is inappropriate to give data about p-level in the abstract.”
We changed the abstract according to your suggestions
Introduction.
“Often some taking points are supported by references to 7 to 10 sources, as if all these publications are about the same thing (lines 55 - 57, 59 - 63). It is necessary to explain in more details exactly what ideas related to this study are contained in these sources. For example, in line 57 there are links to 7 sources (Some possible positive aspects of experiences during the pandemic have also been reported by parents of typically developing children [43-44] and caregivers of children with ASD [26,43,31,40,41,45,46]). And then the detailing of these taking points follows, but without references to sources (lines 63-67). Apparently, the references listed above should be connected with specific details in lines 63-67”.
We connected many references with more specific details.
Instruments.
“There is no data on the Greek adaptation of Parenting Stress Index Short-Form (PSI-SF)”.
We added the Cronbach alpha as calculated in our sample
Results.
“The tables show Statistical significance without taking into account Multiple comparisons problem. It would be necessary to introduce Multiple testing correction, for example, using the Holm or Benjamini-Hochberg. Even though it obviously won't affect the results significantly”.
The correction for multiple comparisons in the analysis was not applied since the comparisons were not made in different groups of the same dependent variable as a post-hoc procedure, but independent variables were compared during follow up. Due to the large number of comparisons presented in the table and the quite small sample size, this correction could lead to a very high rate of false negatives. Also, an extra analysis controlling for confounding factors along with regression analysis has being added.
References
“Violation of numbering in the list of references – double numbering.”
Numbering has been corrected
Lines 315 - 319 - Table 3 is shown twice.
According to another reviewer’s suggestion tables 2-5 were merged.
Round 2
Reviewer 3 Report
My remarks are highlighted in color.
Abstract.
“A brief description of the results should be made not in terms of measurements (variables), but in terms of constructs. And it is inappropriate to give data about p-level in the abstract.”
We changed the abstract according to your suggestions
OK, the changes are accepted.
Introduction.
“Often some taking points are supported by references to 7 to 10 sources, as if all these publications are about the same thing (lines 55 - 57, 59 - 63). It is necessary to explain in more details exactly what ideas related to this study are contained in these sources. For example, in line 57 there are links to 7 sources (Some possible positive aspects of experiences during the pandemic have also been reported by parents of typically developing children [43-44] and caregivers of children with ASD [26,43,31,40,41,45,46]). And then the detailing of these taking points follows, but without references to sources (lines 63-67). Apparently, the references listed above should be connected with specific details in lines 63-67”.
We connected many references with more specific details.
OK, the changes are accepted.
Instruments.
“There is no data on the Greek adaptation of Parenting Stress Index Short-Form (PSI-SF)”.
We added the Cronbach alpha as calculated in our sample
Of course, in order to adapt a multi-scale questionnaire, it is necessary to justify for a new sample: a) factor validity; b) reliability for each scale, and not just the general one. Okay, let's skip factor validity, but it is necessary to specify reliability for each scale.
Results.
“The tables show Statistical significance without taking into account Multiple comparisons problem. It would be necessary to introduce Multiple testing correction, for example, using the Holm or Benjamini-Hochberg. Even though it obviously won't affect the results significantly”.
The correction for multiple comparisons in the analysis was not applied since the comparisons were not made in different groups of the same dependent variable as a post-hoc procedure, but independent variables were compared during follow up. Due to the large number of comparisons presented in the table and the quite small sample size, this correction could lead to a very high rate of false negatives. Also, an extra analysis controlling for confounding factors along with regression analysis has being added.
You didn't understand clearly the wording of the question. We are not talking about multiple comparisons, but about a multitude of simultaneously tested hypotheses. This problem is usually referred to as the false discovery rate (FDR). You can briefly familiarize yourself with this problem at least in Wikipedia, finding the article "False discovery rate". There you will find the ways to solve it and links to other sources. In your case, if you follow Table 2, 17 hypotheses are tested at the same time, so it is necessary to introduce an amendment for a 17-fold test of the hypothesis about the difference of 2 samples. According to my quick calculations, with the use of the Benjamini-Hochberg procedure, the largest p= 0.008, the 7th in ascending order, with the use of B-H- procedure "turns" into p = 0.019. So you're lucky this time but you have to keep that in mind for the future.
References
“Violation of numbering in the list of references – double numbering.”
Numbering has been corrected
OK, the changes are accepted.
Lines 315 - 319 - Table 3 is shown twice.
According to another reviewer’s suggestion tables 2-5 were merged.
OK, the changes are accepted.
Author Response
Thank you for your comments. Below you can find our response.
- Instruments.
“Of course, in order to adapt a multi-scale questionnaire, it is necessary to justify for a new sample: a) factor validity; b) reliability for each scale, and not just the general one. Okay, let's skip factor validity, but it is necessary to specify reliability for each scale.”
We added Cronbach alpha for all scales of PSI-SF
-Results.
“ You didn't understand clearly the wording of the question. We are not talking about multiple comparisons, but about a multitude of simultaneously tested hypotheses. This problem is usually referred to as the false discovery rate (FDR). You can briefly familiarize yourself with this problem at least in Wikipedia, finding the article "False discovery rate". There you will find the ways to solve it and links to other sources. In your case, if you follow Table 2, 17 hypotheses are tested at the same time, so it is necessary to introduce an amendment for a 17-fold test of the hypothesis about the difference of 2 samples. According to my quick calculations, with the use of the Benjamini-Hochberg procedure, the largest p= 0.008, the 7th in ascending order, with the use of B-H- procedure "turns" into p = 0.019. So you're lucky this time but you have to keep that in mind for the future.”
We applied the Benjamini-Hochberg procedure and, as you already mentioned in your comment, all significant comparisons remained statistically significant.
All changes in the text are highlighted in yellow
This manuscript is a resubmission of an earlier submission. The following is a list of the peer review reports and author responses from that submission.
Round 1
Reviewer 1 Report
Thank you for allowing me to review your article. This is timely and relevant to parents of children with ASD. The questions that arose during my review were adequately addressed in the limitations section. I would caution the authors to add in the conclusion section the words "for this sample" and throughout the manuscript where relevant. Otherwise, the wording sounds like the results are being generalized to other samples which, given the multiple limitations, cannot be generalized. I would also invite the authors in future studies, if possible, to include the perspectives of siblings or other family members outside of the primary caregiver.
Reviewer 2 Report
It is really interesting to explore the responses of parents of children with ASD in different crisis, and it is very important for further interventions to improve the responses of parents. I think the topic of the study is very interesting, but how to determine the conclusion of study is based on the crisis background? I mean, the differences between the results in the two time points, perhaps just because of the differences of time or others reasons, but not related to the economic crisis or COVID-19.
I believe that the researchers have done a lot of work, but the study design is not rigorous enough to answer the study topic. If the crisis needs to be discussed, it is suggested to control for other important confounding factors, including the time, the income, and symptoms of children and so on, or to set up a control group in a non-crisis state. Otherwise, it is difficult to consider the global crisis environment as a crisis environment for individual parents of children with ASD.